# Bayesian Multi-Scale Neural Network for Crowd Counting

Abhinav Sagar[*1]

[1]Vrije Universiteit Brussel

## Abstract

Crowd counting is a challenging yet critical task in computer vision with applications ranging from public safety to urban planning. Recent advances using Convolutional Neural Networks (CNNs) that estimate density maps have shown significant success. However, accurately counting individuals in highly congested scenes remains an open problem due to severe occlusions, scale variations, and perspective distortions, where people appear at drastically different sizes across the image. In this work, we propose a novel deep learning architecture that effectively addresses these challenges. Our network integrates a ResNet-based feature extractor for capturing rich hierarchical representations, followed by a down-sampling block employing dilated convolutions to preserve spatial resolution while expanding the receptive field. An upsampling block using transposed convolutions reconstructs the high-resolution density map. Central to our architecture is a novel Perspective-aware Aggregation Module (PAM) designed to enhance robustness to scale and perspective variations by adaptively aggregating multi-scale contextual information. We detail the training procedure, including the loss functions and optimization strategies used. Our method is evaluated on three widely used benchmark datasets—ShanghaiTech, UCF-CC-50, and UCF-QNRF—using Mean Absolute Error (MAE) and Mean Squared Error (MSE) as evaluation metrics. Experimental results demonstrate that our model achieves superior performance compared to existing state-of-the-art methods. Additionally, we incorporate principled Bayesian inference techniques to provide uncertainty estimates along with the crowd count predictions, offering a measure of confidence in the model's outputs.

## 1 Introduction

Crowd counting has garnered significant interest in the computer vision community in recent years due to its broad range of practical applications. These include estimating the number of people in political rallies, public demonstrations, concerts, religious gatherings, and sporting events. Moreover, the underlying methodologies can be adapted to related tasks such as counting cells in microscopic images, vehicles in aerial or satellite imagery, and animals in ecological monitoring. Despite its utility, crowd counting—particularly in highly congested scenes—remains a challenging problem. Two major factors contribute to this difficulty: (1) severe occlusions, clutter, and overlaps between individuals, and (2) large variations in scale and appearance due to perspective distortion, where individuals closer to the camera appear much larger than those farther away.

A wide range of algorithms has been proposed to address these challenges. The dominant paradigm in recent years involves the use of Convolutional Neural Networks (CNNs) combined with density map estimation. These methods predict a continuous density map over the input image, which, when integrated, yields the total object count. Training datasets typically provide only point annotations—often marking the head center of each individual—rather than full object labels or bounding boxes. This sparse annotation presents additional challenges for model learning. Earlier approaches to crowd counting relied on object detection or instance segmentation techniques to identify and count individuals. However, these methods proved inefficient and inaccurate, particularly in dense crowd scenarios, due to their high computational cost and poor performance under occlusion.

To address these limitations, regression-based methods were introduced. These approaches bypassed explicit object detection by learning a direct mapping from image features to a global count value. While this reduced the impact of occlusion and overlapping individuals, it failed to capture spatial information and struggled with varying object scales due to perspective effects. The most successful evolution in this field has been the use of density map estimation techniques. These models generate a density map that not only encodes the presence of individuals but also preserves spatial and scale information. By learning a mapping from the input image to a corresponding density distribution, these methods effectively handle both occlusion and scale variation. As a result, density map-based CNN approaches have become the state-of-the-art standard in modern crowd counting.

Bayesian techniques have emerged as powerful tools in deep learning-based crowd counting, offering not only accurate predictions but also principled uncertainty estimation. Unlike traditional determinis-

---

*Email: abhinav.sagar@vub.be

Proceedings of the 7th Northern Lights Deep Learning Conference (NLDL), PMLR 307, 2026.

tic models that provide point estimates, Bayesian approaches model the predictive distribution, enabling the quantification of both aleatoric uncertainty (inherent data noise) and epistemic uncertainty (model uncertainty due to limited data). This is particularly valuable in high-stakes applications such as surveillance and public safety, where understanding the confidence of a prediction is as important as the prediction itself. In the context of crowd counting, Bayesian neural networks can be implemented using methods such as Monte Carlo Dropout, variational inference, or deep ensembles to approximate posterior distributions. These techniques help mitigate overfitting, improve generalization, and offer uncertainty-aware predictions that can be leveraged for downstream tasks like active learning, anomaly detection, or dynamic resource allocation in real-time systems. Incorporating Bayesian inference thus adds a crucial layer of reliability and interpretability to modern crowd counting models.

## 2 Related Work

Several important contributions have advanced the field of crowd counting using deep learning techniques. One of the pioneering works in this domain is by [41], which introduced a cross-scene crowd counting approach using a switchable learning strategy that simultaneously optimized two objectives: crowd density estimation and overall count regression. Building on this idea, [25] proposed an end-to-end trainable switching CNN architecture that automatically selects the most suitable regressor for different crowd regions, improving robustness across varying densities. The concept of using multi-column architectures to capture features at different receptive fields was popularized by [43], who replaced fully connected layers with $1 \times 1$ convolutional layers to reduce parameters while maintaining spatial resolution. Similarly, [1] combined deep and shallow CNNs to capture both low-level and high-level features, enhancing performance in scenes with significant scale variation and occlusion.

[21] introduced an iterative refinement approach where one CNN estimates a coarse density map, which is then progressively refined in a second stage. Meanwhile, [31] proposed a multi-task cascaded CNN that jointly learns crowd count classification and density map regression, allowing shared feature learning and improved generalization. Further innovations include the multi-scale contextual encoding approach of [16], which explicitly models perspective distortion and demonstrates the benefit of multi-scale features in handling scale variation. [42] proposed a scale-adaptive fusion method that concatenates features extracted at different resolutions, while [27] integrated local and global contextual information for predicting counts at multiple levels.

To incorporate top-down scene semantics, [24] employed a feedback mechanism to refine predictions based on global scene context. [29] tackled the perspective challenge using perspective maps encoded as adaptive weighting layers to combine density predictions at multiple scales. [9] further explored scale adaptation with a multi-scale encoder and multi-path decoder framework for high-fidelity density map generation. Hybrid approaches have also emerged. [14] fused detection and regression using an attention mechanism to switch between the two paradigms based on crowd density. [15] introduced a local pattern consistency loss, improving fine-grained density estimation through region-level correlation modeling. Attention mechanisms were also employed in [4] to enable both global and local scale selection via soft attention.

Semi-supervised and unsupervised learning approaches have shown promise as well. [26] utilized an autoencoder to extract transferable features from unlabeled data, while [3] focused on identifying pixel-level subregions with high prediction errors to guide learning. [33] introduced a hierarchical attention framework, combining spatial and global attention modules across multiple scales to enhance focus on relevant crowd regions. Recent research has increasingly focused on integrating uncertainty modeling into crowd counting. [19] and [17] independently proposed Bayesian formulations for crowd counting that yield both point estimates and uncertainty quantification. Their models are capable of estimating both epistemic uncertainty (related to model confidence) and aleatoric uncertainty (inherent data noise), improving reliability in ambiguous or high-density scenarios. [8] presented a unified composition loss that jointly supervises count, density, and localization tasks, pushing the boundary of multi-task learning in dense scenes.

We summarize our main contributions as follows:

- We propose a novel deep neural network architecture for crowd counting, built upon a ResNet-based feature extractor. Our model incorporates a downsampling module using dilated convolutions to preserve spatial resolution, and an upsampling module using transposed convolutions to reconstruct high-quality density maps.

- We introduce a novel Perspective-aware Aggregation Module (PAM) that improves robustness to scale and perspective distortions by adaptively fusing multi-resolution features across the network.

- We provide comprehensive implementation details, including the network architecture, optimization strategy, loss functions, and the evaluation protocol. Our method is evaluated on

three widely used benchmarks—ShanghaiTech, UCF-CC-50, and UCF-QNRF—using standard MAE and MSE metrics.

- Our model achieves state-of-the-art accuracy while significantly reducing the number of parameters compared to existing methods. Furthermore, we incorporate principled Bayesian inference to estimate both epistemic and aleatoric uncertainties, making our system more interpretable and reliable for real-world deployment.

# 3  Proposed Method

## 3.1  Dataset

Experimental evaluations are conducted using three widely used crowd counting datasets: ShanghaiTech part A and part B, UCF-CC 50, and UCF-QNRF. These datasets are described as follows:

- ShanghaiTech is made up of two datasets labeled as part A and part B. In Part A, there are 300 images for training and 182 images for testing, while Part B has 400 training images and 316 testing images. Most of the images are of very crowded scenes, such as rallies and large sporting events. Part A has a significantly higher density than Part B.

- UCF-CC-50 contains 50 gray images with different resolutions. The average count for each image is 1,280, and the minimum and maximum counts are 94 and 4,532, respectively.

- UCF-QNRF is the third dataset used in this work, which has 1535 images with 1.25 million point annotations. It is a challenging dataset because it has a wide range of counts, image resolutions, light conditions, and viewpoints. The training set has 1,201 images, and 334 images are used for testing.

## 3.2  Network Architecture

Our proposed network architecture consists of three primary modules: a feature extraction block, a reconstruction (upsampling) block, and a multi-head prediction module for density estimation and uncertainty quantification.

The feature extraction block is built upon a modified ResNet backbone enhanced with dilated (atrous) convolutions, which serve as the downsampling mechanism. Unlike traditional max-pooling or stride-based downsampling, dilated convolutions allow the receptive field to expand without sacrificing spatial resolution. This design is particularly effective in crowd-counting scenarios where objects (i.e., people) appear at varying scales due to perspective

distortion. By capturing multi-scale contextual information, the dilated ResNet-based encoder mitigates issues related to severe occlusion and scale variation.

Following the encoder, the reconstruction or upsampling block utilizes transposed convolutional layers (also known as deconvolutions) to progressively restore the spatial resolution of the feature maps. To preserve fine-grained details lost during encoding, skip connections are introduced between corresponding encoder and decoder layers. These lateral connections form a U-Net-like structure, facilitating efficient gradient flow and enabling the network to fuse low-level and high-level information.

The final part of the architecture is the multi-head output module, which consists of three branches:

- The density map head, which produces a high-resolution density map. When integrated spatially, this map yields the total estimated count of people in the input image.

- The epistemic uncertainty head, which estimates uncertainty arising from model limitations, is approximated via the Monte Carlo dropout technique.

- The aleatoric uncertainty head, which models noise inherent in the input data, is particularly relevant in cluttered or poorly illuminated scenes.

An overview of the architecture, including the layer-wise structure, is illustrated in Figure 1.

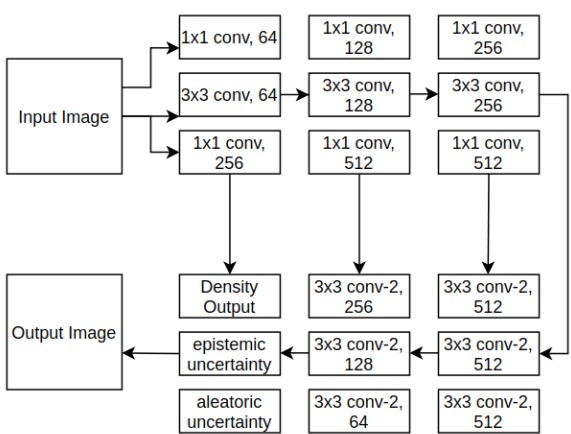

**Figure 1.** Illustration of our proposed network architecture. In the diagram, 1×1 and 3×3 denote convolutional filter sizes, 64, 128, 256 indicate the receptive field sizes (feature channel depths), conv represents dilated convolutional layers used in the downsampling path, and conv-2 denotes transposed convolutional layers used for upsampling in the reconstruction path.

This carefully designed architecture enables our model to effectively estimate crowd density while simultaneously quantifying uncertainty in a principled Bayesian framework.

## 3.3 Optimization

While training the network, the vanishing gradient problem showed up, ie weights of the connections were turning out to be zero. To alleviate this, instance normalization was used after both convolutional and transposed convolutional layers as defined below:

$$y = ReLU\left(\sum_{i=0}^{d} w_i \cdot ReLU\left(\gamma_i \cdot \frac{x_i - \mu_i}{\sqrt{\sigma_i^2 + \epsilon}} + \beta_i\right) + b\right) \quad (1)$$

Where $w$ and $b$ are the weight and bias terms of the convolution layer, $\gamma$ and $\beta$ are the weight and bias terms of the Instance Normalization layer, $\mu$ and $\sigma$ are the mean and variance of the input.

Previous works have used multi-column architecture [43] to deal with the various scales at which objects might be present in the image. The problem with these methods is that the number of columns gives a direct measure of the scale at which they can recognize individual objects. To tackle this, we propose a new technique to aggregate the filters with sizes 1×1, 3×3, and 5×5. ReLU is applied after every convolutional and transposed convolutional layer. The filter branches make our network robust and can be extended by using more filters to tackle crowd counting in dense scenes. Our aggregation modules stacked on top of each other behave as ensembles, thus minimizing overfitting, which is a challenge with deep networks. The novel aggregation module used in our work is shown in Figure 2:

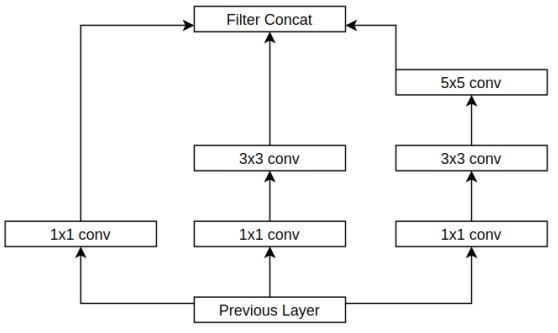

**Figure 2.** Illustration of our aggregation module.

## 3.4 Loss Function

Most existing work uses pixel-wise Euclidean loss for training the network. This gives a measure of estimation error at the pixel level, which is defined below:

$$L_E = \frac{1}{N}\|F(X,\theta) - Y\|^2 \quad (2)$$

where $\theta$ denotes a set of the network parameters, $N$ is the number of pixels in density maps, $X$ is the input image and $Y$ is the corresponding ground truth density map, $F(X,\theta)$ denotes the estimated density map.

We also incorporate the SSIM index in our loss to measure the deviation of the prediction from the ground truth. The SSIM index is used in image quality assessment. It computes similarity between two images from three local statistics, i.e., mean, variance, and covariance. The range of SSIM values is from -1 to 1, and the SSIM is equal to 1 when the two images are identical. The SSIM index is defined in:

$$SSIM = \frac{(2\mu_F\mu_Y + C_1)(2\sigma_{FY} + C_2)}{(\mu_F^2 + \mu_Y^2 + C_1)(\sigma_F^2 + \sigma_Y^2 + C_2)} \quad (3)$$

where $C_1$ and $C_2$ are small constants to avoid division by zero. The next term of the loss function can be written by averaging over the integral, as shown below:

$$L_S = \frac{1}{N}\sum_x SSIM(x) \quad (4)$$

Where $N$ is the number of pixels in the density maps. $L_S$ gives a measure of the difference between the network predictions and ground truth. The final loss function, by adding the two terms, can be written as shown in Equation 5:

$$L_{tot} = \alpha_E L_E + \alpha_S L_S \quad (5)$$

where $\alpha_E$ and $\alpha_S$ are constants. In our experiments, we set both $\alpha_E$ and $\alpha_S$ as 0.5 to give equal weights to both the terms.

## 3.5 Evaluation Metrics

For crowd counting, the count error is measured by two metrics, Mean Absolute Error (MAE) and Mean Squared Error (MSE), which are commonly used for quantitative comparison. These metrics are defined as in Equation 6 and Equation 7:

$$MAE = \frac{1}{N}\sum_{i=1}^{N}\left|C_i - C_i^{GT}\right| \quad (6)$$

$$MSE = \sqrt{\frac{1}{N}\sum_i^{N}\left|C_i - C_i^{GT}\right|^2} \quad (7)$$

Where $N$ is the number of test samples, $C_i$ and $C_i^{GT}$ are the estimated and ground truth count corresponding to the $i^{th}$ sample, which is given by the

integration of the density map. MAE shows the accuracy of the predicted result, while MSE measures the robustness of the prediction.

## 3.6 Uncertainty Estimation

In predictive modeling, especially for safety-critical tasks like crowd counting in highly congested scenes, quantifying uncertainty is essential. Uncertainty provides a measure of confidence in model predictions and is typically categorized into two main types: epistemic uncertainty and aleatoric uncertainty.

Epistemic uncertainty (also known as model uncertainty) arises due to the lack of sufficient knowledge or data. It reflects uncertainty in the model parameters and can, in theory, be reduced by collecting more diverse and representative training data. Epistemic uncertainty is especially prominent in regions of the input space that the model has not seen during training.

Aleatoric uncertainty (also called data uncertainty) stems from inherent noise in the observations—for example, occlusion, scale ambiguity, low-resolution imagery, poor lighting, or clutter. This form of uncertainty cannot be eliminated by gathering more data, as it is intrinsic to the data-generating process.

To capture epistemic uncertainty, we leverage Bayesian Neural Networks (BNNs), where the weights of the network are treated as distributions rather than deterministic point estimates. This is achieved by placing a prior distribution over the weights and approximating the posterior using Monte Carlo dropout technique. This probabilistic treatment allows the model to express uncertainty in the learned representations, especially useful in out-of-distribution or ambiguous regions.

On the other hand, aleatoric uncertainty is modeled directly as a learnable component of the network's output, allowing the model to predict heteroscedastic noise—i.e., noise that varies across input samples. To simultaneously learn the predictive mean and variance, we define a loss function that captures both types of uncertainty. The loss function used for training our network is depicted below:

$$\mathcal{L}(\theta) = \frac{1}{D} \sum_i \frac{1}{2\sigma^2} \|y_i - \hat{y}_i\|^2 + \frac{1}{2} \log \sigma^2 \quad (8)$$

where $y_i$ is the $i^{th}$ pixel of the output density $y$ corresponding to input $x$ and $D$ is the number of output pixels. Note that the observation noise $\sigma^2$ captures how much noise is present in the outputs, and it stays constant for all data points.

## 3.7 Algorithm

Let input images be denoted by $\{x_n\}_{n=1}^N$ and ground truth images by $\{y_n\}_{n=1}^N$. The trainable parameters

for the network are denoted by $\theta, \phi$, which are obtained from a uniform distribution $\{1, \ldots, K\}$. The model parameters are denoted by $\theta$ for the shared backbone and $\phi$ for task-specific heads (density map, epistemic uncertainty, aleatoric uncertainty). To capture the predictive uncertainty, we model $\theta$ and $\phi$ as random variables and approximate their distributions by sampling from a uniform prior over $K$ different weight samples during training.

The complete algorithm used in our work is shown below:

---

**Algorithm 1:** Bayesian Multi-Scale Neural Network for Crowd Counting

---

**1** Require: Input images $\{x_n\}_{n=1}^N$, GT images $\{y_n\}_{n=1}^N$

**2** Initialize parameters $\theta, \phi$

**3** **for** *each epoch* **do**

**4**      **for** *n = 1 to N* **do**

**5**          Sample $\theta, \phi \sim$ Uniform $\{1, \ldots, K\}$

**6**          Compute predictions $[y_n] = f_{\theta_k}(x_n)$

**7**          Calculate loss:

            $L(\theta) = \frac{1}{D} \sum_i \frac{1}{2\sigma^2} \|y_i - \hat{y}_i\|^2 + \frac{1}{2} \log \sigma^2$

**8**          Update $\theta_k$ using gradient descent

            $\frac{dL(\theta_k)}{d\theta_k}$

**9**      **end**

**10** **end**

---

The uniform sampling from $K$ different parameterizations introduces stochasticity to emulate Bayesian posterior sampling. The loss function jointly minimizes prediction error and learns to predict aleatoric uncertainty. Epistemic uncertainty is captured through weight sampling at inference time by averaging multiple forward passes.

# 4 Experimental Results and Analysis

## 4.1 Quantitative Results

To evaluate the effectiveness of our proposed Bayesian multi-scale network for crowd counting, we conducted extensive experiments on three benchmark datasets: ShanghaiTech, UCF-CC 50, and UCF-QNRF. Our model consistently achieves the lowest Mean Absolute Error (MAE) and Mean Squared Error (MSE) across all datasets, demonstrating both accuracy and robustness. In addition, we report the number of trainable parameters to show that our method is not only accurate but also highly efficient.

The ShanghaiTech dataset consists of two subsets: Part A with dense crowd scenes and Part B with sparse crowds. As shown in Table 1, our method outperforms most previous state-of-the-art approaches

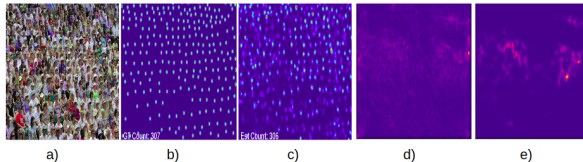

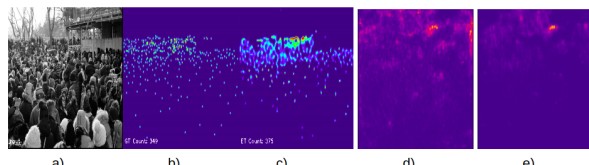

**Figure 3.** Qualitative results on the ShanghaiTech dataset. Each row shows: (a) Input image, (b) Ground truth density map, (c) Predicted density map, (d) Epistemic uncertainty, and (e) Aleatoric uncertainty.

**Figure 4.** Qualitative results on the UCF-QNRF dataset. Each row shows: (a) Input image, (b) Ground truth density map, (c) Predicted density map, (d) Epistemic uncertainty, and (e) Aleatoric uncertainty.

in terms of both MAE and MSE on both subsets. Compared to CSRNet, CP-CNN, and Switch-CNN, our model achieves better accuracy while maintaining a significantly smaller parameter footprint (see Table 4).

UCF-CC 50 is a highly challenging dataset due to its extremely limited size (only 50 images) and wide density variation. Table 2 shows that our method achieves the lowest MSE while maintaining a competitive MAE, outperforming several high-capacity models such as SFCN and DUBNet. The balance between accuracy and generalization under extreme data scarcity demonstrates the strength of our Bayesian modeling approach.

UCF-QNRF is one of the largest and most diverse crowd counting datasets, with highly congested scenes and large image resolution. Table 3 highlights that our method achieves the best MSE performance and remains competitive in MAE compared to other strong baselines like DUBNet and CAN. Our improved uncertainty modeling helps in better generalization across such diverse scenes.

Beyond accuracy, our model is designed to be lightweight and computationally efficient. As shown in Table 4, our method achieves state-of-the-art performance using only 0.24 million parameters, which is significantly fewer than even the most compact prior works like SANet (0.91M). This makes our architecture well-suited for deployment on edge devices and real-time applications.

## 4.2 Qualitative Results and Uncertainty Analysis

Figures 3 and 4 present qualitative results of our proposed method on representative samples from the ShanghaiTech and UCF-QNRF datasets, respectively. Each row displays a crowd image from the test set alongside five visualizations: the input image, the ground-truth density map, the predicted density map, and the corresponding epistemic and aleatoric uncertainty maps.

The epistemic uncertainty reflects the model's uncertainty due to limited training data or model capacity. It is learned through multiple stochastic forward passes and captures the spread of the model's predictions. On the other hand, the aleatoric uncertainty

represents the inherent noise and ambiguity in the input data—such as motion blur, low resolution, or occlusion—which cannot be reduced even with more data.

In both datasets, we observe the following:

- Higher uncertainty in regions of dense crowds: In areas with high object overlap or extreme perspective distortion, both epistemic and aleatoric uncertainty values are notably elevated. This is expected, as accurately estimating density in such regions is inherently more difficult.

- Correlation between uncertainties: There is a visible spatial alignment between regions of high epistemic and aleatoric uncertainty, indicating that ambiguous regions in the image (e.g., occluded or poorly lit people) challenge the model both from a data and modeling perspective.

- Sharper epistemic patterns in sparse areas: In low-density regions, the epistemic uncertainty tends to capture specific regions where the model is unsure about the existence of crowd presence, while aleatoric uncertainty remains low—highlighting model doubt rather than data ambiguity.

- The color intensity in the uncertainty maps—particularly the red regions—correlates with the degree of uncertainty: more red denotes higher uncertainty. This visual feedback can be crucial in real-world applications where knowing the model's confidence is as important as the prediction itself.

In summary, the proposed method not only provides accurate density maps but also delivers meaningful uncertainty quantification that helps interpret the reliability of its outputs, especially under challenging crowd scenes.

## 5 Conclusions

In this work, we proposed a novel deep learning framework for crowd counting that integrates accurate density estimation with robust uncertainty quantification. The architecture is built upon a

**Table 1.** Comparison with state-of-the-art methods on ShanghaiTech dataset (lower is better). Left: Part A, Right: Part B

| Method | MAE (A) | MSE (A) | MAE (B) | MSE (B) |
|---|---|---|---|---|
| Zhang et al. [41] | 181.8 | 277.7 | 32.0 | 49.8 |
| MCNN [43] | 110.2 | 173.2 | 26.4 | 41.3 |
| Cascaded-MTL [31] | 101.3 | 152.4 | 20.0 | 31.1 |
| Switch-CNN [25] | 90.4 | 135.0 | 21.6 | 33.4 |
| CP-CNN [32] | 73.6 | 106.4 | 20.1 | 30.1 |
| CSRNet [13] | 68.2 | 115.0 | 10.6 | 16.0 |
| SANet [2] | 67.0 | 104.5 | 8.4 | 13.6 |
| SFCN [38] | 64.8 | 107.5 | 7.6 | 13.0 |
| CAN [16] | **62.3** | 100.0 | 7.8 | 12.2 |
| DUBNet [19] | 64.6 | 106.8 | 7.7 | 12.5 |
| **Ours** | 63.2 | **95.6** | **7.3** | **10.6** |

**Table 2.** Comparison with state-of-the-art methods on UCF-CC 50 dataset (lower is better)

| Method | MAE | MSE |
|---|---|---|
| MCNN [40] | 377.6 | 509.1 |
| Cascaded-MTL [31] | 322.8 | 397.9 |
| Switch-CNN [25] | 318.1 | 439.2 |
| D-ConvNet [30] | 288.4 | 404.7 |
| L2R [37] | 279.6 | 388.9 |
| CSRNet [13] | 266.1 | 397.5 |
| ic-CNN [21] | 260.9 | 365.5 |
| SANet [2] | 258.4 | 334.9 |
| SFCN [38] | 214.2 | 318.2 |
| CAN [16] | **212.2** | 243.7 |
| DUBNet [19] | 243.8 | 329.3 |
| **Ours** | 216.7 | **225.1** |

**Table 3.** Comparison with state-of-the-art methods on UCF-QNRF dataset (lower is better)

| Method | MAE | MSE |
|---|---|---|
| MCNN [40] | 277 | 426 |
| Cascaded-MTL [31] | 252 | 514 |
| Switch-CNN [25] | 228 | 445 |
| CSRNet [13] | 135.5 | 207.4 |
| SFCN [38] | **102.0** | 171.4 |
| CAN [16] | 107 | 183 |
| DUBNet [19] | 105.6 | 180.5 |
| **Ours** | 106.7 | **165.1** |

ResNet-based feature extractor, augmented with dilated convolutions in the downsampling path to preserve spatial resolution and capture multi-scale context. The upsampling path leverages transposed convolutions, while skip connections between corresponding encoder and decoder layers promote effective feature reuse, mitigate vanishing gradients, and help prevent overfitting. To enhance prediction robustness, we introduced a feature aggregation module that facilitates rich semantic fusion across different levels of the network. Furthermore, the network branches into three output heads: a density map for crowd count estimation, and two auxiliary heads to estimate epistemic and aleatoric uncertainty, thereby making the model's predictions more interpretable and trustworthy. We also detailed the Bayesian learning framework employed to model epistemic uncertainty via variational weight sampling and used a log-likelihood-based loss function to capture aleatoric noise. A complete training algorithm was provided to demonstrate how uncertainty-aware optimization is implemented end-to-end. Experimental evaluations on three benchmark datasets—ShanghaiTech, UCF-CC 50, and UCF-QNRF—demonstrate that our model achieves state-of-the-art performance, consistently outperforming prior methods in both MSE and MAE metrics. Additionally, our model achieves this with a significantly lower parameter count, showcasing its efficiency and scalability. Importantly, the integration of uncertainty modeling addresses the black-box nature of traditional deep neural networks by providing pixel-wise estimates of prediction confidence. This capability is especially crucial for deployment in high-stakes real-world applications such as public safety, event monitoring, and urban planning.

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

**Table 4.** Comparison of model size in terms of parameter count (in millions). Lower is better.

| Method | Switch-CNN [25] | CP-CNN [32] | CSRNet [13] | SANet [2] | **Ours** |
|---|---|---|---|---|---|
| **Parameters (M)** | 15.11 | 68.4 | 16.26 | 0.91 | **0.24** |

[3] Zhi-Qi Cheng, Jun-Xiu Li, Qi Dai, Xiao Wu, and Alexander G Hauptmann. Learning spatial awareness to improve crowd counting. In *Proceedings of the IEEE International Conference on Computer Vision*, pages 6152–6161, 2019.

[4] Mohammad Hossain, Mehrdad Hosseinzadeh, Omit Chanda, and Yang Wang. Crowd counting using scale-aware attention networks. In *2019 IEEE Winter Conference on Applications of Computer Vision (WACV)*, pages 1280–1288. IEEE, 2019.

[5] Yaocong Hu, Huan Chang, Fudong Nian, Yan Wang, and Teng Li. Dense crowd counting from still images with convolutional neural networks. *Journal of Visual Communication and Image Representation*, 38:530–539, 2016.

[6] Siyu Huang, Xi Li, Zhongfei Zhang, Fei Wu, Shenghua Gao, Rongrong Ji, and Junwei Han. Body structure aware deep crowd counting. *IEEE Transactions on Image Processing*, 27 (3):1049–1059, 2017.

[7] Haroon Idrees, Imran Saleemi, Cody Seibert, and Mubarak Shah. Multi-source multi-scale counting in extremely dense crowd images. In *Proceedings of the IEEE conference on computer vision and pattern recognition*, pages 2547–2554, 2013.

[8] Haroon Idrees, Muhmmad Tayyab, Kishan Athrey, Dong Zhang, Somaya Al-Maadeed, Nasir Rajpoot, and Mubarak Shah. Composition loss for counting, density map estimation and localization in dense crowds. In *Proceedings of the European Conference on Computer Vision (ECCV)*, pages 532–546, 2018.

[9] Xiaolong Jiang, Zehao Xiao, Baochang Zhang, Xiantong Zhen, Xianbin Cao, David Doermann, and Ling Shao. Crowd counting and density estimation by trellis encoder-decoder networks. In *Proceedings of the IEEE Conference on Computer Vision and Pattern Recognition*, pages 6133–6142, 2019.

[10] Di Kang and Antoni Chan. Crowd counting by adaptively fusing predictions from an image pyramid. *arXiv preprint arXiv:1805.06115*, 2018.

[11] Diederik P Kingma and Jimmy Ba. Adam: A method for stochastic optimization. *arXiv preprint arXiv:1412.6980*, 2014.

[12] Shohei Kumagai, Kazuhiro Hotta, and Takio Kurita. Mixture of counting cnns: Adaptive integration of cnns specialized to specific appearance for crowd counting. *arXiv preprint arXiv:1703.09393*, 2017.

[13] Yuhong Li, Xiaofan Zhang, and Deming Chen. Csrnet: Dilated convolutional neural networks for understanding the highly congested scenes. In *Proceedings of the IEEE conference on computer vision and pattern recognition*, pages 1091–1100, 2018.

[14] Jiang Liu, Chenqiang Gao, Deyu Meng, and Alexander G Hauptmann. Decidenet: Counting varying density crowds through attention guided detection and density estimation. In *Proceedings of the IEEE Conference on Computer Vision and Pattern Recognition*, pages 5197–5206, 2018.

[15] Lingbo Liu, Zhilin Qiu, Guanbin Li, Shufan Liu, Wanli Ouyang, and Liang Lin. Crowd counting with deep structured scale integration network. In *Proceedings of the IEEE International Conference on Computer Vision*, pages 1774–1783, 2019.

[16] Weizhe Liu, Mathieu Salzmann, and Pascal Fua. Context-aware crowd counting. In *Proceedings of the IEEE Conference on Computer Vision and Pattern Recognition*, pages 5099–5108, 2019.

[17] Zhiheng Ma, Xing Wei, Xiaopeng Hong, and Yihong Gong. Bayesian loss for crowd count estimation with point supervision. In *Proceedings of the IEEE International Conference on Computer Vision*, pages 6142–6151, 2019.

[18] Mahdi Maktabdar Oghaz, Anish R Khadka, Vasileios Argyriou, and Paolo Remagnino. Content-aware density map for crowd counting and density estimation. *arXiv preprint arXiv:1906.07258*, 2019.

[19] Min-hwan Oh, Peder A Olsen, and Karthikeyan Natesan Ramamurthy. Crowd counting with decomposed uncertainty. In *AAAI*, pages 11799–11806, 2020.

[20] Daniel Onoro-Rubio and Roberto J López-Sastre. Towards perspective-free object counting with deep learning. In *European Conference on Computer Vision*, pages 615–629. Springer, 2016.

[21] Viresh Ranjan, Hieu Le, and Minh Hoai. Iterative crowd counting. In *Proceedings of the European Conference on Computer Vision (ECCV)*, pages 270–285, 2018.

[22] David Ryan, Simon Denman, Sridha Sridharan, and Clinton Fookes. An evaluation of crowd counting methods, features and regression models. *Computer Vision and Image Understanding*, 130:1–17, 2015.

[23] Abhinav Sagar and RajKumar Soundrapandiyan. Semantic segmentation with multi scale spatial attention for self driving cars. *arXiv preprint arXiv:2007.12685*, 2020.

[24] Deepak Babu Sam and R Venkatesh Babu. Top-down feedback for crowd counting convolutional neural network. In *Thirty-second AAAI conference on artificial intelligence*, 2018.

[25] Deepak Babu Sam, Shiv Surya, and R Venkatesh Babu. Switching convolutional neural network for crowd counting. In *2017 IEEE Conference on Computer Vision and Pattern Recognition (CVPR)*, pages 4031–4039. IEEE, 2017.

[26] Deepak Babu Sam, Neeraj N Sajjan, Himanshu Maurya, and R Venkatesh Babu. Almost unsupervised learning for dense crowd counting. In *Proceedings of the AAAI Conference on Artificial Intelligence*, pages 8868–8875, 2019.

[27] Chong Shang, Haizhou Ai, and Bo Bai. End-to-end crowd counting via joint learning local and global count. In *2016 IEEE International Conference on Image Processing (ICIP)*, pages 1215–1219. IEEE, 2016.

[28] Zan Shen, Yi Xu, Bingbing Ni, Minsi Wang, Jianguo Hu, and Xiaokang Yang. Crowd counting via adversarial cross-scale consistency pursuit. In *Proceedings of the IEEE conference on computer vision and pattern recognition*, pages 5245–5254, 2018.

[29] Miaojing Shi, Zhaohui Yang, Chao Xu, and Qijun Chen. Revisiting perspective information for efficient crowd counting. In *Proceedings of the IEEE Conference on Computer Vision and Pattern Recognition*, pages 7279–7288, 2019.

[30] Zenglin Shi, Le Zhang, Yun Liu, Xiaofeng Cao, Yangdong Ye, Ming-Ming Cheng, and Guoyan Zheng. Crowd counting with deep negative correlation learning. In *Proceedings of the IEEE conference on computer vision and pattern recognition*, pages 5382–5390, 2018.

[31] Vishwanath A Sindagi and Vishal M Patel. Cnn-based cascaded multi-task learning of high-level prior and density estimation for crowd counting. In *2017 14th IEEE International Conference on Advanced Video and Signal Based Surveillance (AVSS)*, pages 1–6. IEEE, 2017.

[32] Vishwanath A Sindagi and Vishal M Patel. Generating high-quality crowd density maps using contextual pyramid cnns. In *Proceedings of the IEEE International Conference on Computer Vision*, pages 1861–1870, 2017.

[33] Vishwanath A Sindagi and Vishal M Patel. Ha-ccn: Hierarchical attention-based crowd counting network. *IEEE Transactions on Image Processing*, 29:323–335, 2019.

[34] Nitish Srivastava, Geoffrey Hinton, Alex Krizhevsky, Ilya Sutskever, and Ruslan Salakhutdinov. Dropout: a simple way to prevent neural networks from overfitting. *The journal of machine learning research*, 15(1):1929–1958, 2014.

[35] Dmitry Ulyanov, Andrea Vedaldi, and Victor Lempitsky. Instance normalization: The missing ingredient for fast stylization. *arXiv preprint arXiv:1607.08022*, 2016.

[36] Jennifer Vandoni, Emanuel Aldea, and Sylvie Le Hégarat-Mascle. Evaluating crowd density estimators via their uncertainty bounds. In *2019 IEEE International Conference on Image Processing (ICIP)*, pages 4579–4583. IEEE, 2019.

[37] Jia Wan, Wenhan Luo, Baoyuan Wu, Antoni B Chan, and Wei Liu. Residual regression with semantic prior for crowd counting. In *Proceedings of the IEEE Conference on Computer Vision and Pattern Recognition*, pages 4036–4045, 2019.

[38] Qi Wang, Junyu Gao, Wei Lin, and Yuan Yuan. Learning from synthetic data for crowd counting in the wild. In *Proceedings of the IEEE conference on computer vision and pattern recognition*, pages 8198–8207, 2019.

[39] Mingliang Xu, Zhaoyang Ge, Xiaoheng Jiang, Gaoge Cui, Pei Lv, Bing Zhou, and Changsheng Xu. Depth information guided crowd counting for complex crowd scenes. *Pattern Recognition Letters*, 125:563–569, 2019.

[40] Lingke Zeng, Xiangmin Xu, Bolun Cai, Suo Qiu, and Tong Zhang. Multi-scale convolutional neural networks for crowd counting. In *2017 IEEE International Conference on Image Processing (ICIP)*, pages 465–469. IEEE, 2017.

[41] Cong Zhang, Hongsheng Li, Xiaogang Wang, and Xiaokang Yang. Cross-scene crowd counting via deep convolutional neural networks. In

*Proceedings of the IEEE conference on computer vision and pattern recognition*, pages 833–841, 2015.

[42] Lu Zhang, Miaojing Shi, and Qiaobo Chen. Crowd counting via scale-adaptive convolutional neural network. In *2018 IEEE Winter Conference on Applications of Computer Vision (WACV)*, pages 1113–1121. IEEE, 2018.

[43] Yingying Zhang, Desen Zhou, Siqin Chen, Shenghua Gao, and Yi Ma. Single-image crowd counting via multi-column convolutional neural network. In *Proceedings of the IEEE conference on computer vision and pattern recognition*, pages 589–597, 2016.

