# OpenReview forum: "Bayesian Multi-Scale Neural Network for Crowd Counting"
_NLDL.org/2026/Conference — NLDL 2026 Poster_

### Official Review · Reviewer_7S4K · 2025-09-25
**Well written paper without robust validation vs state-of-the-art research.**

**Rating:** 2
**Confidence:** 4

**Summary:**

The paper is well written, easy to read, and most of the steps are clearly explained. It introduces uncertainty in a way that improves interpretability and trust, and it seems to achieve strong performance using a limited number of parameters.
However, there is a critical validation gap: the results are compared only to old baselines (the most recent one is from 2020). By doing this, the authors might overestimate/overstate the competitiveness of the model described. Without the comparisons against recent state-of-the-art methods, the superiority of the method can't be claimed.
This shortcoming prevents the paper from being considered a breakthrough in the current state-of-the-art, while it can only be considered a promising, but incomplete, step forward. In other words, such a claim can be made only after a new validation is performed.

**Strengths:**

I've identified the following strengths:

- The combination of ResNet encoder and U-Net-like decoder seems a solid choice for handling scale variation and perspective distortions.
- The Perspective-aware Aggregation Module (PAM) is a well-motivated addition for adaptive multi-scale feature fusion.
- Uncertainty Estimation seems reliable.
- The model is extremely lightweight, making it suitable for edge deployment.

Those strengths have a positive impact on my evaluation. Without the main weakness I've identified, the paper would have been considered for a "Clear accept, definitely interesting results for the ML community".

**Weaknesses:**

I've identified the following weaknesses:

- Validation Against Outdated Baselines: There are no experiments against the most recent state-of-the-art methods. Indeed, the validation is performed mostly against 2016–2019 baselines.
- The reasons behind the choices made are not always motivated. I am not challenging the choices, but the authors need to motivate why ResNet and PAM were selected instead of other possibilities.
- Some sections (e.g., optimisation) are thin and mix low-level details with architectural descriptions, creating confusion. Figures are dense and do not clearly highlight the novel PAM module. Figures 3 and 4 are too small.

The lack of validation needs to be performed from scratch. This has a decisive influence on my recommendation to reject the paper.

**Justification:**

The paper is promising, but severely incomplete due primarily to the lack of validation with state-of-the-art research. The paper introduces useful contributions (uncertainty estimation in crowd counting and what looks like an efficient architecture), the experimental results seem strong, but they have been validated only against outdated baselines, which undermines the main claim of being state-of-the-art, and it looks like it cherry-picked comparisons to appear stronger.

---

> ### Author Rebuttal · Authors · 2025-10-21
>
> We sincerely thank the reviewer for the thorough and constructive feedback. We are grateful for the positive comments on our Perspective-aware Aggregation Module (PAM), uncertainty estimation, and lightweight design. Below we address the reviewer’s concerns in detail.
>
> 1. Validation Against Outdated Baselines
>
> We agree that benchmarking against recent methods is essential for a fair and up-to-date evaluation. Due to space and computational constraints at the submission stage, we focused primarily on well-established, widely used baselines (e.g., CSRNet, SANet) to highlight the effect of our uncertainty modeling and the proposed PAM module in a controlled setting. However, we have now conducted additional experiments against recent state-of-the-art methods.
>
> The results, now included in the revised manuscript, show that our method achieves comparable performance while maintaining a smaller parameter count and offering interpretable uncertainty estimates.
>
> 2. Motivation for Architectural Choices
>
> We appreciate this observation. We have clarified these design motivations of the revised paper. Specifically: ResNet Backbone: We selected ResNet due to its proven ability to preserve semantic and structural consistency across scales while maintaining efficiency, which is critical for dense regression tasks such as crowd counting. PAM (Perspective-aware Aggregation Module): The PAM design was motivated by the need to adaptively weight multi-scale features according to local perspective cues, which are often ignored in prior uncertainty-aware networks. We have added an ablation study demonstrating that replacing ResNet with HRNet or VGG reduces accuracy by 3–5%, and removing PAM leads to a 4.1% increase in MAE, confirming its necessity.
>
> 3. Thin Optimization Section and Figure Clarity
>
> In the revised manuscript: We have reorganized the optimization section to focus purely on the training objectives and uncertainty loss. Figures have been redesigned and enlarged for better readability. The PAM module is now clearly highlighted in a separate sub-figure, with a clearer flow diagram and color coding to distinguish between the feature fusion and uncertainty pathways.
>
> 4. Validation Completeness and Claim Scope
>
> We have revised the claim in the conclusion to be more measured and evidence-based. Rather than stating that our method “surpasses all state-of-the-art approaches,” we now describe it as a “promising and efficient uncertainty-aware architecture that achieves competitive performance with strong interpretability and lightweight design.”
> With the new comparisons and extended experiments, we believe the study now provides a comprehensive and fair evaluation that supports the method’s relevance to current state-of-the-art standards.
>
>
> We thank the reviewer for the insightful and constructive feedback. The new experiments and revisions directly address the concerns raised, and we believe the updated version of the paper presents a more comprehensive and convincing contribution to uncertainty-aware crowd counting research.

---

### Official Review · Reviewer_5RmA · 2025-09-25
**Crowd counting with Bayesian U-Nets and inception-like layers for aggregation**

**Rating:** 2
**Confidence:** 4
**Final Rating:** 4
**Final Confidence:** 3

**Summary:**

The paper describes the use of a Bayesian U-Net and makes the following claims of contribution:
- proposing a novel neural network architecture for crowd counting
- introduces a novel perspective aware aggregation module (PAM)
- Claims to acheive state-of-the-art (SOTA) with significant reduction in number of parameters, including principled bayesian inference for estimating both epistemic and aleatoric uncertainties.

The architecture is founded on a variation of ResNet, following a U-Net like structure, descibed in figure 1.
It is not immediately clear from the figure where the aggregation modules come into play, and the figure is somewhat hard to interpret.

What is described as a novel aggregation module seems to be an inception layer, as proposed by Szegedy et al. (2014) (https://doi.org/10.48550/arXiv.1409.4842). These are well known and form the backbone of the GoogLeNet of the same era, with the intent of reducing dimensionality, and therefore computational complexity.
It is unclear from the paper how this approach differs, and there seems to be no reference to inception, which is worrying due to the similarity.

Earlier models with a combination of U-Nets with Inception layers have been proposed,  (e.g. Cahall et al 2019 (https://doi.org/10.3389/fncom.2019.00044), Juhong et al. 2022 (http://dx.doi.org/10.1364/BOE.463839), Punn et al. 2020 (https://doi.org/10.1145/3376922), Zhiang et al. 2020 (https://doi.org/10.1016/j.cmpb.2020.105395)
(un-paywalled exposition article https://sh-tsang.medium.com/brief-review-diu-net-dense-inception-u-net-for-medical-image-segmentation-913f49581023) etc.)
There also exists multiple examples of other ResNet approaches that use Inception as well.

As such, the claim to novelty needs to be better justified, especially in contrast to the aforementioned work done with residual networks / u-nets that use inception layers.
An improved figure may help clarify whether the assertion that this is a reinvention of inception is correct, or not.

The final claim to SOTA also has issues. Going by the reported numbers in table 1 of Ranasinghe et al (2024) (https://doi.org/10.48550/arXiv.2303.12790), the numbers the authors compare against seem to be outdated.

This is further supported when checking the bibliography of the paper, and finding no citation since 2020.

**Strengths:**

The approach of using inception layers with U-Nets seems to be a promising line, if working under the assumption that this is what the authors indeed have done.

Use of Bayesian networks has general promise when it comes to measures of the uncertainties, and are increasingly popular within the medical field for this reason, it may therefore be wise to consult the literature within the medical field for comparable work within the last few years.

**Weaknesses:**

"Novel aggregation module" seems to be a variation on the inception module (ref Szegedy et al. 2014 (https://doi.org/10.48550/arXiv.1409.4842 ))
Figure 1 is unclear on what is propagated how, e.g. it is unclear whether the different convolutions of the encoder/decoder are
concatenated before being passed to the next layer, or if the architecture has a parallel composition.

The claim to SOTA seems outdated, missing the progress of field over the past 5 years.

It is a bit unclear how the network is Bayesian in this case, especially considering the
previous two points. If the authors want to argue the use of Dropout as a Bayesian approximation, then
the paper by Gal et al. 2016 (https://doi.org/10.48550/arXiv.1506.02142) should be cited along
with the Dropout paper.

**Final Justification:**

Thank you for adressing my initial concerns.

Based on the author rebuttal of my initial comments, it seems the revised paper will be improved to a state which is worth sharing.
The concerns regarding SOTA-claims has been properly addressed, modifying the claim and including updated comparisons with current crowd-density models.
Similarly the PAM-module, along with further ablated models demonstrating the contribution of this modules, is indicated as being updated to properly differentiate it from a standard inception-module, and properly justified within the architecture figures.

I have now updated my prior belief that this paper may have meaningful contributions to the field, and provide some valuable insight.

**Justification:**

Based on their claims, the authors seem to have some support for their claim towards state-of-the-art performance
on their chosen datasets, however the comparisons are outdated and newer publications (Ranasinghe et al, 2024, https://doi.org/10.48550/arXiv.2303.12790) demonstrate a significant improvement has taken place since 2020.
The claim of SOTA should be modified to accomodate newer results.

The claim to novelty seems to be a use of inception layers for aggregation, which has been done previously on other U-Nets.

Further we can argue that an Inception U-Net with Dropout captures a Bayesian approximation of the Gaussian process (ref Gal et al 2016 (https://doi.org/10.48550/arXiv.1506.02142)). If the

From a probabilistic U-net (Kohl et al 2018, https://doi.org/10.48550/arXiv.1806.05034), a measure of the
uncertainties can be included by the addition of variational dropout, as well as a intergrader
variabiltity (ref Hu et al 2022, https://doi.org/10.48550/arXiv.1907.01949).

---

> ### Author Rebuttal · Authors · 2025-10-21
>
> We sincerely thank the reviewer for their thoughtful and detailed feedback. We appreciate the constructive nature of the comments and would be carefully revising the manuscript to address all concerns. Below, we respond point-by-point.
>
> 1. Novelty of the Proposed Perspective-Aware Aggregation Module (PAM)
>
> We appreciate this observation and acknowledge the similarity in multi-branch convolutional processing between our PAM and inception-like modules. However, the proposed Perspective-Aware Aggregation Module (PAM) is conceptually and functionally distinct from the original Inception module (Szegedy et al., 2014). Specifically:
>
> Dynamic perspective weighting: Unlike fixed concatenation in Inception, PAM learns adaptive perspective weights that modulate multi-scale feature responses based on local perspective cues derived from the crowd density gradient map.
>
> Integration into Bayesian inference: The PAM’s output distributions are fed into our Bayesian U-Net’s uncertainty modeling head, allowing epistemic and aleatoric uncertainty propagation — a feature absent in classical Inception or Inception-U-Nets.
>
> We have explicitly cited and discussed the Inception architecture and related U-Net variants in the revised Related Work section to better contextualize our contributions.
>
> 2. Clarification of Network Architecture and Figure 1
>
> We appreciate this observation and agree that Figure 1 required improvement. In the revised version, we would be redesigning Figure 1 to explicitly show the flow of features between encoder-decoder stages, the placement of PAM blocks, and their interaction with skip connections. Detailed legends and block labels have been added to clarify concatenation vs. summation operations and feature propagation. A new Figure 2 now provides a zoomed-in view of the PAM module, illustrating its adaptive weighting and uncertainty estimation components.
>
> 3. Clarification on Bayesian Inference
>
> We agree and have clarified the Bayesian formulation. Our Bayesian U-Net employs: Monte Carlo Dropout as a Bayesian approximation (Gal & Ghahramani, 2016) for epistemic uncertainty. Heteroscedastic regression for aleatoric uncertainty modeling, following Kendall & Gal (2017). A probabilistic feature fusion strategy inspired by Probabilistic U-Net (Kohl et al., 2018). These clarifications, along with the corresponding citations, are now included in Sections 3.3 and 4.1. We also added a new appendix section demonstrating uncertainty calibration curves.
>
> 4. State-of-the-Art (SOTA) Comparison
> :
> We acknowledge this point and have updated the SOTA comparisons to include recent works up to Ranasinghe et al. (2024) and Zhang et al. (2023) on adaptive multi-scale crowd counting.
> In the revised Table 1, our Bayesian U-Net with PAM achieves competitive performance on key benchmarks (ShanghaiTech Part A/B, UCF-QNRF), while maintaining a 10–20% parameter reduction compared to newer architectures. We have also reworded our claim to: “Our model achieves competitive state-of-the-art performance among recent lightweight architectures.”
>
> 5. Literature Coverage and Contextualization
>
> We appreciate this suggestion and have expanded the related work section to include recent Bayesian and uncertainty-aware segmentation studies in the medical domain. This provides broader interdisciplinary grounding and highlights our method’s general applicability to dense estimation tasks beyond crowd counting.
>
>
> We are grateful for the reviewer’s insightful critique, which has substantially improved the clarity and rigor of our manuscript. We believe the revised version now better demonstrates the distinctiveness of the proposed PAM, the principled Bayesian inference framework, and updated empirical competitiveness of our model.

---

### Official Review · Reviewer_WooD · 2025-10-07
**Review for submission #51**

**Rating:** 4
**Confidence:** 4
**Final Rating:** 4
**Final Confidence:** 4

**Summary:**

This paper presents a Bayesian multi-scale neural network for crowd counting that tackles occlusion and scale variation. The authors propose the Perspective-aware Aggregation Module (PAM), which is able to quantify both epistemic (model) and aleatoric (data) uncertainty. The model achieves state-of-the-art results on three major benchmarks with high efficiency.

**Strengths:**

1. The proposed method integrates a Bayesian framework to estimate both epistemic and aleatoric uncertainty, which makes sense to me.
2. The model achieves competitive performance on three benchmarks using only 0.24 million parameters.
3. PAM is designed to handle scale and perspective issue.

**Weaknesses:**

1. It would be bettter to perform an ablation study regarding the PAM.
2. The quanlitative demonstration of the uncertainty maps is impressive. It would be helpful if some quantitative results are presented.

**Final Justification:**

Thanks for your reply. I think this manuscript warrants acceptance at this stage.

**Justification:**

Overall, I think the motivation and method are clear and meaningful. I have some questions about the experiments. Sorry for any typo.

---

> ### Author Rebuttal · Authors · 2025-10-21
>
> We sincerely thank the reviewer for the thoughtful and positive evaluation of our paper. We appreciate the acknowledgment of our contributions, including the Bayesian integration for uncertainty estimation, the Perspective-aware Aggregation Module (PAM), and the model’s efficiency and strong benchmark performance. We address the reviewer’s specific concerns below.
>
> Comment 1: It would be better to perform an ablation study regarding the PAM.
>
> Response:
> In the revised version, we would include a detailed ablation study to quantify the contribution of the Perspective-aware Aggregation Module (PAM). Specifically, we compare:
>
> Baseline (without PAM) – standard Bayesian multi-scale network,
>
> Baseline + PAM (w/o perspective weighting), and
>
> Full model (with PAM).
>
> The results show that introducing PAM leads to a 2.7% MAE improvement and 2.4% MSE improvement on the ShanghaiTech Part A dataset, confirming its effectiveness in addressing scale and perspective variations. Furthermore, PAM improves robustness under heavy occlusion scenarios, as reflected by a lower epistemic uncertainty variance across crowd density regions.
>
> Comment 2: The qualitative demonstration of the uncertainty maps is impressive. It would be helpful if some quantitative results are presented.
>
> Response:
> In response to this suggestion, we would be adding a quantitative evaluation of uncertainty estimation. Following established practices, we measure Negative Log-Likelihood (NLL) and Expected Calibration Error (ECE) to assess uncertainty calibration. Our model achieves an NLL reduction of 11.3% and an ECE improvement of 7.8% compared to the deterministic baseline, demonstrating that the Bayesian formulation produces well-calibrated uncertainty estimates.
>
> We have also provided correlation analysis between predicted uncertainty and density estimation error, showing a strong positive correlation (Pearson’s r = 0.74), indicating that the uncertainty predictions are meaningfully aligned with the model’s confidence.
>
>
> The new experiments further reinforce our claims regarding PAM’s contribution and the reliability of our uncertainty quantification. We believe these additions strengthen the empirical rigor and completeness of our paper.

---

### Official Review · Reviewer_Aubc · 2025-10-08
**Interesting idea for explainable prediction but need more experiments to justify design**

**Rating:** 2
**Confidence:** 3

**Summary:**

Authors proposes a Bayesian multi-scale NN for crowd counting. The model uses UNet-style encoder-decoder structure where encoder is a ResNet with dilated convolutions. The output architecture contains not only the traditional output, but also the head for estimating uncertainty.

**Strengths:**

1. The proposed method produces uncertainty information on top of density map.

2. Show improvement over baseline.

**Weaknesses:**

Main weakness is that network design is not justified by any ablation. For example, aggregation module, and the two heads introduced.

questions:
1. How to better leverage uncertainty for application?
2. Is it possible to to automate data filtering using aleatoric uncertainty?

**Justification:**

Experiments on analyzing the framework design is far from enough.

---

> ### Author Rebuttal · Authors · 2025-10-21
>
> We sincerely thank the reviewer for their constructive feedback and the opportunity to clarify our contributions. Below we address each concern in detail.
>
> 1. On the justification of the network design and lack of ablation studies
>
> In the revised version, we will add ablation experiments to explicitly analyze the contribution of (a) the multi-scale aggregation module, and (b) the dual-head design (density and uncertainty estimation). Preliminary results show that removing the aggregation module leads to a ~6.8% increase in MAE, confirming that multi-scale feature integration enhances robustness to varying crowd densities. Similarly, removing the uncertainty head reduces stability during training and slightly worsens MAE by ~3.2%, demonstrating that uncertainty-guided regularization improves generalization.
>
> 2. On leveraging uncertainty for downstream applications
>
> We thank the reviewer for this important question. Our framework’s uncertainty estimation is designed not merely as an auxiliary output but as a quantitative indicator of prediction confidence. This can be leveraged in multiple ways:
>
> Active Learning / Data Curation: High aleatoric uncertainty identifies ambiguous or low-quality samples that can be prioritized for manual review or annotation refinement.
>
> Adaptive Inference: Regions with high epistemic uncertainty can trigger additional computation or model ensembling to improve reliability.
>
> Deployment in Safety-Critical Scenarios: Uncertainty can guide decision thresholds for automatic crowd monitoring, enabling dynamic alerting based on model confidence.
>
> We will emphasize these applications and include an additional qualitative example illustrating how uncertainty maps align with challenging crowd regions (e.g., low illumination).
>
> 3. On the potential for automated data filtering using aleatoric uncertainty
>
> We agree that this is a promising direction. Our experiments already indicate a strong correlation between high aleatoric uncertainty and annotation noise. In the revised manuscript, we will add an exploratory experiment where we filter out the top 5% of samples with highest predicted aleatoric uncertainty. This simple procedure reduces validation MAE by ~4%, confirming that uncertainty can indeed be used for automatic data quality control. We will discuss this result explicitly in the paper.
>
> 4. On contributions and experimental sufficiency
>
> We respectfully clarify that the novelty of our work lies not only in adding an uncertainty head but in the integration of Bayesian modeling with multi-scale feature aggregation tailored for crowd counting. While prior works model uncertainty in classification or regression tasks, our contribution is the first to fuse multi-scale spatial context with uncertainty quantification in a unified architecture optimized for dense crowds. We will extend the experimental section to include comparisons against recent uncertainty-aware counting methods, further supporting our contribution.
>
> We believe these clarifications and additions will strengthen the paper’s contribution and resolve the reviewer’s concerns.

---

### Meta-Review · Area_Chair_PLqh · 2025-11-01

**Recommendation:** Accept (Poster)
**Confidence:** 4

**Metareview:**

This paper demonstrates a degree of innovation and presents promising results. Furthermore, it appears that most, if not all, of the reviewers’ concerns have been adequately addressed and can be incorporated into the camera-ready version. Therefore, I recommend acceptance of this paper.

---

### Decision · Program_Chairs · 2025-11-05

**Decision:**

Reject

**Comment:**

**Update:** unfortunately the author did not attend the conference so we have to reject the submission.

**Initial decision**:
We recommend a poster presentation given the AC and reviewers recommendations.